# Alternative Splicing of Human Telomerase Reverse Transcriptase (hTERT) and Its Implications in Physiological and Pathological Processes

**DOI:** 10.3390/biomedicines9050526

**Published:** 2021-05-09

**Authors:** Anna A. Plyasova, Dmitry D. Zhdanov

**Affiliations:** Institute of Biomedical Chemistry, Pogodinskaya st 10/8, 119121 Moscow, Russia; annalyasova13@gmail.com

**Keywords:** alternative splicing, telomerase, splice variants, human telomerase reverse transcriptase (hTERT), telomeres, lymphocytes, endonuclease G, apoptosis

## Abstract

Alternative splicing (AS) of human telomerase catalytic subunit (hTERT, human telomerase reverse transcriptase) pre-mRNA strongly regulates telomerase activity. Several proteins can regulate AS in a cell type-specific manner and determine the functions of cells. In addition to being involved in telomerase activity regulation, AS provides cells with different splice variants that may have alternative biological activities. The modulation of telomerase activity through the induction of hTERT AS is involved in the development of different cancer types and embryos, and the differentiation of stem cells. Regulatory T cells may suppress the proliferation of target human and murine T and B lymphocytes and NK cells in a contact-independent manner involving activation of TERT AS. This review focuses on the mechanism of regulation of hTERT pre-mRNA AS and the involvement of splice variants in physiological and pathological processes.

## 1. Introduction

The stability of chromosomes at their 3′ ends is supported by telomere (TTAGGG) DNA repeats. Recombination-based alternative lengthening of telomeres and elongation of telomeres by telomerase are two mechanisms for maintaining telomere length in cells with unlimited replicative potential or immortalized cells, including stem cells, germ cells, activated lymphocytes, endothelial cells, and cancer cells [1,2,3,4,5]. hTERT and hTR comprise the catalytic core of telomerase, whereas the holoenzyme contains additional species-specific accessory proteins. The most frequent mechanism for telomere elongation among various cells is the functioning of telomerase, a ribonucleoprotein complex consisting of two key subunits: human telomerase RNA (hTR), which acts as a transcription template for newly synthetized telomeres, and human telomerase reverse transcriptase (hTERT), whose enzymatic activity controls the grade of telomerase activity. Although hTR is constitutively expressed in most tissues in human cells, hTERT expression is highly regulated at the transcriptional and posttranscriptional levels. However, the suggestion that hTERT levels regulate telomerase activity only applies to most somatic tissues, as opposed to the brain, where hTR is downregulated very early during development and is thus most likely responsible for the disappearance of the activity [6].

Transcriptional regulation of hTERT was extensively studied, and the chromatin environment, DNA methylation, DNA looping, promoter mutations, and the binding of transcription factors were shown to affect the strength of gene expression [7,8].

Most genes of higher eukaryotes have an interrupted structure in certain coding regions—exons alternate with noncoding sequences and introns. Gene transcription leads to the formation of pre-mRNA, a molecule that has both exons and introns. Primary pre-mRNA transcripts undergo modifications before being translated, i.e., capping at the 5′- end, and the synthesis of the polyA sequence at the 3′-end of the transcript. The second event in the maturation of mRNA is a splicing of exons and excision of introns. Post-transcriptional maturation of pre-mRNA plays an essential role in providing the biodiversity of protein products encoded by a single gene due to the process of alternative splicing (AS) of pre-mRNA. In this process, particular exons, or parts of exons, may be included within or excluded from the final matured mRNA. Consequently, the proteins translated from alternatively spliced pre-mRNAs will contain differences in their amino acid sequence, and often in their biological functions too [9]. To date, several types of AS are described for human pre-mRNAs (as illustrated in Figure 1A). Exon skipping is the most common mode in mammalian pre-mRNAs. In this case, an exon may be spliced out of the primary transcript or retained. Mutually exclusive exons are a mode when only one of two exons is retained in mRNAs after splicing, but not both. Alternative donor/acceptor sites are two modes when an alternative 5′ or 3′ splice junction is used. The rearrest mode in mammals is an intron retention when a sequence may be spliced out as an intron or simply retained. This is distinguished from exon skipping because the retained sequence is not flanked by introns [10,11]. In addition to these primary modes, AS may provide alternative poly-AA(A)n site, an alternative site for the beginning of translation (start-codon) or the formation of preliminary stop-codon [12]. Pre-mRNA splicing is conducted by a multiprotein complex spliceosome (as illustrated in Figure 1B), which consists of six main subunits (U1, U2, U2AF, U4, U5, and U6) and about 300 other proteins [13]. The first step of spliceosome functioning is the interaction of its subunits with the splice sites: U1 binds to the 5′site, U2 binds to the branch site, and U2AF binds to the polypyrimidine site of the 3′ splice site. The second step is the joining of the U4, U5, and U6 subunits to U1 and U2AF, which leads to the convergence of the 5′-and 3′ - splice sites. At the third step, the intron is removed in two stages. At the first stage, a transesterification reaction occurs, in which the adenosine of the branch site binds to the guanidine of the 5′ site’s GU sequence. At the second stage, the exons are ligated, and the nucleotides of the branched intron are removed in the form of lariat. An exon may be constitutive (always included in the mRNA) or alternative (may be included or excluded) to splice variants. The usage of a splice site may be enhanced or suppressed by its proximity to local cis-regulatory sequences (as illustrated in Figure 1C), such as exonic splicing enhancers (ESEs), exonic splicing silencers (ESSs), intronic splicing enhancers (ISEs), and intronic splicing silencers (ISSs) [14,15]. The cis-regulatory sequences are in turn bound by trans-acting factors or splicing factors. Most often they are serine-arginine-repeat proteins (SR proteins). The spliceosome performs the deletion of intron and exons ligation during splicing, while the functioning of SR proteins is crucial for the determination of sites that will be deleted or retained.

The regulation of telomerase activity and the alternative functions of hTERT through its pre-mRNA processing were less investigated. AS is one of the mechanisms that mediates the diversity of mRNAs encoded by a single gene. Precursor message RNA (pre-mRNA) subjected to different AS events determines the fate of mature mRNA variants. Splice variants of mRNA may be translated to variants of proteins that have functions that are frequently different from the full-length form, or in cases of altered capacity for translation, such splice variants are degraded by nonsense-mediated decay [16]. The AS mechanism is a tissue-specific process, and the formation of mature splice variants of a given pre-mRNA depends on the repertoire of splicing-regulatory proteins and on features of spliceosome regulation [17]. Regulation of telomerase through AS of hTERT pre-mRNA is considered to be a cell-specific process, as the presence of different hTERT splice variants is in good agreement with the grade of telomerase activity in normal cells and in cells from pathological tissues [8]. The exact regulatory mechanisms of hTERT AS and the biological functions of splice variants was not elucidated. Some recent data suggested novel features for the mechanisms of hTERT pre-mRNA AS and the roles of its splice variants in telomerase regulation alongside other cellular functions of hTERT in normal and pathological processes.

## 2. Regulatory Mechanisms for hTERT Pre-mRNA AS

Since hTERT is the major subunit maintaining telomerase activity, its expression and activity are highly regulated at many levels, including promoter core region organization, protein folding, post-translational modification, and interaction partners [7]. Regulation of hTERT functions by AS of its pre-mRNA is thought to be crucial because even small amounts of TERT may have significant cellular consequences [18]. While significant amounts of knowledge about TERT and telomerase functioning came from the study of model organisms [19], the following data will refer to the regulation of AS of human TERT unless otherwise stated.

### 2.1. Alternative Splice Variants of hTERT

The human hTERT gene (42 kb) is located on chromosome 5p15.33 and spans 16 exons and 15 introns. The full-length hTERT transcript encodes an active 1132-amino acid (127-kDa) protein. Bioinformatics and mutational studies collectively established that hTERT contains four main structural elements (as illustrated in Figure 2A): (i) a long telomerase essential N-terminal extension (TEN domain); (ii) conserved DNA- and TERT RNA-binding domains (TRBD); (iii) a central catalytic reverse-transcriptase (RT) domain; and (iv) a short C-terminal extension (CTE domain). The TEN domain is important for the appropriate action of telomerase at telomeres, as mutations in the DAT (dissociates activities of telomerase) region abolish telomerase processivity but not catalytic activity in vivo [20]. The TEN domain contains the BH3-like motif 135WGLLLRRVGDDVLVHL152, a short peptide sequence found in BCL-2 family proteins, and interacts with the antiapoptotic BCL-2 family proteins MCL-1 and BCL-xL, suggesting a functional link between hTERT and the mitochondrial pathway of apoptosis [21].

The TRBD domain has several conserved telomerase-specific motifs, including the CP, QFP, and TS motifs that are important for TERT-TR binding interactions and the grade of template copying during telomere synthesis [22,23]. The TEN domain and TRBD cooperate to ensure sequence-specific TERT-TR binding interactions and optimal template positioning during telomerase assembly and/or telomere synthesis. The RT domain contains five conserved RT motifs that are responsible for the catalytic activity of the enzyme [24]. The CTE domain is responsible for different protein-protein interactions and regulates the cellular localization of the protein [25].

Activation of hTERT is primarily regulated by the phosphorylation of protein kinase C (PKC) isoenzymes, thereby enhancing telomerase activity [26,27]. hTERT contains a nuclear localization signal (NLS) consisting of two clusters of basic amino acids (^222^RRR^224^ and ^236^KRPRR^240^) [28]. Additionally, putative Akt phosphorylation sites are located at ^220^GARRRGGSAS^229^ and ^817^AVRIRGKSYV^826^, and the phosphorylation of serine 227 (but not serine 824) is required for nuclear translocation of hTERT [28,29,30]. Extranuclear localization of hTERT is regulated by the phosphorylation of Src tyrosine kinase at tyrosine 707 [31].

To date, 22 different splice variants of hTERT mRNA were identified, which are combinations of five deletions and two insertion splice events (as illustrated in Figure 2B–D) [32,33,34,35]. Only wild-type, the full-length hTERT mRNA with neither deletions nor insertions, encodes a protein that can assemble into an active telomerase holoenzyme [36,37]. In telomerase-positive cells, the full-length hTERT splice variants are expressed in the range of 1–90%, and their expression level determines the grade of telomerase activity [38,39,40].

The most studied hTERT splice variants originate from deletion events during mRNA maturation and are likely to be cell- or tissue-specific. The skipping of 36 nucleotides (nt) in the cryptic 3′ splice-acceptor site (i.e., the alpha region of splicing) at the beginning of exon 6 results in an alpha minus (α‒) variant, which is in the canonical reading frame. The hTERT α‒ variant has a deletion of 12 amino acids (aa) in the RT motif and acts as a dominant-negative protein that can bind to hTR but cannot maintain telomeres [41]. This truncated variant is rather abundant in cancer cells [38] and human activated lymphocytes [42].

The skipping of 182 nt in the beta splicing region results in the deletion of exons 7 and 8, and the shift of the reading frame leads to the formation of a premature stop codon in exon 10 and causes an early termination of translation [38,41]. Such a beta minus (β‒) splice variant is mostly degraded by nonsense-mediated decay; however, in certain cancer cells, this variant interacts with polyribosomes and is translated into a truncated protein, which is not able to form active telomerase but protects cells from apoptosis [37]. Some cells may express this β‒ hTERT variant in human activated CD4^+^ T lymphocytes in their steady state, and its induction by endonuclease EndoG leads to the inhibition of telomerase [43]. The expression levels of the β-variant vary between cells in the range of 10–90%. Combinations of alpha and beta splice variants (α+β−and α−β+) have also been detected in cells and can range within 1–15% depending on cell type. The biological significance of these variants for cells remains to be investigated, as they must be degraded by nonsense-mediated decay.

Another in-frame splicing event occurs in the gamma splicing region and results in the skipping of 189 nt of exon 11. Gamma minus (γ–) hTERT has an affected RT domain (missing conserved 63 aa from the catalytic core of the protein) and acts as a dominant-negative protein when it is expressed at sufficient levels [44]. Although γ-deletion hTERT showed low expression (up to 2% of total hTERT) in cell lines with high telomerase activity, this variant may occasionally be detected in samples with low telomerase activity, such as hepatocellular carcinoma nodules. Combinations of γ– splice variants such as α-γ–, β-γ–, and α-β-γ– were also detected at low levels in telomerase-positive cells; however, their functions still require investigation [44].

Withers et al. [39] demonstrated that the human epidermoid carcinoma cell line A431 expresses the hTERT splice variant lacking full exon 2 (delta 2, ∆2 variant), and the skipping of 1354 nt results in frameshift and translation termination in exon 3. Exon 2 encodes a part of the TRB domain, and the variant that lacks it would be unable to bind hTR. The authors demonstrated that this truncated protein is being translated as a 12-kDa peptide, and in a study with higher precision, it was shown that there are approximately 20–40 molecules of ∆2 hTERT mRNA copies per cell. No functions of the ∆2 variant were investigated; thus, its biological role remains unknown [10].

In a study by Hrdličková et al. [34], eight novel hTERT splice variants were identified in a screen of telomerase-positive and telomerase-negative cell lines. Four of these variants, lacking either a part of exon 2, all of exon 2, exons 2 through 8, or exons 2 through 13 (Δ2p, Δ2, Δ2–8, and Δ2–13), contain premature stop codons. Two other variants retain the second exon but lack the RT domain. One splice variant contains a deletion-encompassing exons 4 through 13 (Δ4–13) and retains the original hTERT reading frame. The second variant lacks a part of exon 3 and all of exons 4 to 12 (Δ3p-12), which introduces a premature stop codon. These variants lack regions of hTERT where the previously described α- and β- splicing events occur. Finally, one additional variant (Δ4C) lacks the 4th exon and contains an insertion of parts of the 3rd and 4th introns. This variant maintains the original reading frame. Variant Δ4–13 is expressed in two lines of dividing telomerase-negative U2-OS cells (human osteosarcoma cells) and Saos-2 (human osteosarcoma), which both employ recombination-based lengthening of telomeres, and in telomerase-positive HeLa (human cervix carcinoma) cells in culture. The authors evaluated whether this variant may be involved in cell proliferation by performing the noncanonical functions (not associated with telomere maintenance) of telomerase. In both cell lines, the overexpression of Δ4–13 resulted in an increase in cell proliferation without an increase in telomerase activity.

Several intron retention splice variants were also described. Sequencing of cDNA plasmid clones from five different tissues, lung tumor and adjacent tissue, colon tumor, and K562 and HL60 cell lines, revealed a number of hTERT splice variants that have insertions [35]. Three variants involve the insertion of sequences from intron 2. The variant Ins-i2 (1–389) inserted the first 349 nucleotides of intron 2 and results from the use of an alternative 5′-splice donor sequence within intron 2. The same splice site was also used to produce the variant Ins-i2 (82–349). In this variant, an additional splicing event removed the first 181 nucleotides of intron 2 by use of the normal 5′-splice donor sequence and an alternative 3′-splice acceptor sequence, leaving nucleotides 182–349 of intron 2 in the spliced product. The third variant involving intron 2, Ins-i2 (?–5273), results from the use of an alternative 5′-splice donor sequence within intron 2, but the precise characteristics of this splice variant were not determined. The remaining variants contain inserted sequences from intron 14. The variants Ins-i14 (623–705) and Ins-i14 (623–703) both result from a double splicing event in intron 14, which leaves an internal fragment of intron 14 (nucleotides 623–705 and 623–703, respectively) in the spliced product. As shown, many of these hTERT splice variants have premature stop codons and must be degraded. However, two variants with insertions are subjected to translation and act as dominant negatives to suppress telomerase activity. The first variant, INS3, contains a 159-nt (nucleotides 622–781) insertion from the end of intron 14, which encodes an additional 44 aa. The insertion results in a premature stop codon in exon 15 [45]. The variant INS4 contains a 600-nt insertion of intron 14, encoding 17 aa, followed by a stop codon. The expression of INS3 and INS4 is tissue-specific and, when expressed, may account for 1–15% of the total hTERT mRNAs [46,47]. Finally, a variant (INTR1) that retains the first intron and contains a premature stop codon was also identified; however, no information about its function is available [34].

It must be considered that TERT pre-mRNA from different species is also subjected to AS. Although some transcriptional TERT patterns are identical to those in humans, most patterns are species-specific. To date, mRNA splice variants different from hTERT were described for chickens [34], rats and mice [48], primates [39], worms [49], and dogs [50], among other species.

### 2.2. Regulation of hTERT Pre-mRNA AS

Similar to most AS, hTERT pre-mRNA is processed by the major splicing machinery, the spliceosome [51], which is regulated by both intronic/exonic elements (cis-elements) close to splice sites and long-range interactions [52] (as illustrated in Figure 3). The most extensively studied are the events that switch hTERT splicing patterns between the two most abundant variants, which are the full-length and β- variants.

Wong et al. [52], using hTERT minigene constructs, found three regulatory sequences that are responsible for the formation of the β- variant. First, a block of 26 short repeats of 38-nt sequences located in intron 6 (termed variable number tandem repeat (VNTR) or block 6 (B6) repeats). Second, a direct repeat within intron 6 (DR6), which consists of 256 nt. Third, a direct repeat within intron 8 (DR6), which consists of 285 nt. These sequences demonstrated 85% homology among species, and the number of B6 repeats varied from 18–38 repeats among different individuals. Later, it was shown that B6 is required for β- splicing, but DR6 and DR8 are not sufficient to skip exons 7 and 8. Moreover, the intronic location of these elements rather than their sequences could determine AS [53]. Several other VNTR sites and DR6- and DR8-related sequences (termed Alu elements) were found in hTERT introns, which provide a provision for another hTERT AS.

The following studies identified a number of trans-factors and their cis-sequences that determine the choice of splice variants. It was demonstrated that β- splicing is controlled by SR proteins [37]. SRSF11 promoted the deletion of exons 7 and 8, while hnRNPH2 and hnRNPL were responsible for the formation of the full-length variants. RNA secondary structures may sterically provide occlusion, exposure, or approximation of cis-elements [54], which can enhance the formation of β- variants. By modeling the hTERT mRNA secondary structure, Wong M.S. et al. [55] demonstrated the possibility of approaching exon 6 5′ and exon 9 3′ splice sites, thereby promoting exon skipping. The results suggested that a minimum of nine 38-bp repeats is necessary for RNA:RNA pairing in hTERT pre-mRNA to change the proximity of exon 6 and 9 splice junctions and/or expose the necessary docking sites for SR proteins or the spliceosome for splice site selection.

The search of binding sites for SR proteins identified 2 sites in introns 6 and 8, sites upstream of, and in, exon 9 that matched the consensus sequences (AAGAA, AAUAA or AACAA) [56]. Interestingly, some hnRNPH2-binding sites overlapped with SRSF11-binding sites. As inclusion or exclusion of β- depends on whether exon 6 is joined to exon 7 or exon 9, respectively, use of the 3′ splice site of either intron 6 or 8 is central to this AS decision. Together, these data suggest that SRSF11 and hnRNPH2 compete for binding to these sites to stimulate either β- site exclusion or inclusion. Additionally, SRSF2 binding motifs located in the 3′ end of intron 6 were shown to regulate β- deletion [57].

Xiao et al. [58] proposed an unveiled function of RBM10 that regulates hTERT splicing by binding to the GGU motif of pancreatic cancer cells to repress the production of the full-length hTERT. Loss of RBM10 promotes cell proliferation, invasion, and xenograft growth. The GGU motif was reported to be a consensus RBM10 binding site, and in vivo RNA immunoprecipitation assays confirmed that RBM10 was recruited to the sites next to the 5′ splice sites of hTERT introns 7 and 8. Site-specific mutagenesis (GGU to GAU) within the RBM10 binding sites repressed the RBM10-mediated skipping of exons 7 and 8.

In the series of studies by Ludlow et al. [59,60], a novel potent regulator neuro-oncological ventral antigen 1 (NOVA1) was identified by bioinformatic approaches, and its ability to inhibit the deletion of exons 7 and 8 was confirmed in knockout experiments. Active site (YCAY, where Y = C; or U) x7 for NOVA1 is located within the previously mentioned DR8 region, and binding to NOVA1 results in its function as a splicing factor. Another possible mechanism is that NOVA1 may regulate the upstream transcription factors of hTERT cells and promote the full-length expression of hTERT [61]. Subsequent experiments demonstrated that NOVA1 binds to pre-mRNA in tandem with polypyrimidine tract binding protein 1 (PTBP1) [60]. The knockdown of PTBP1 in cancer cells resulted in the downregulation of the full-length variant and a reduction of telomerase activity in lung cancer cells.

The study of the role of the apoptotic endonuclease EndoG in the regulation of β- AS in hTERT revealed two novel active sites located at the 5′ end of intron 8, which are sensitive to SpR20 and SPr40 splicing factors, and their activity can be modulated by specific splice-switching oligonucleotides [62]. The mechanism by which EndoG modulates AS of hTERT pre-mRNA is thoroughly reviewed in the next section.

Several other factors can also influence hTERT splicing, but the mechanisms are still far from being understood. Depletion of the chromatin remodeling protein Brm in NCI-H1299 cells led to a decrease in hTERT α+β+ and β- variants. The proposed mechanism relies on the observation that Brm and the splicing factors PSF and p54/(nrb)/NONO can bind to the hTERT gene close to exon 7, indicating the possibility of cotranscriptional splicing [63]. TGF-β1 could downregulate c-MYC and subsequently decrease the expression of the full-length hTERT in human skin keratinocytes by retaining high levels of the inactive β- variant. This result suggests a novel mechanism for TGF-β1-mediated regulation of telomerase [64]. Moreover, several potential cis-elements were predicted in exon 5 to exon 9 of hTERT by bioinformatic analysis [65]. The identification of these potential exonic splicing elements of hTERT might be helpful for the design of antisense oligonucleotides, which could modulate AS of hTERT pre-mRNA and consequently biological properties of hTERT protein.

Although the previously mentioned studies shed light on the regulation of β- AS, studies describing the mechanisms of induction of other splice variants are lacking.

### 2.3. Modulation of hTERT Pre-mRNA AS by Endonuclease G

The first proposal that apoptotic endonuclease G (EndoG) may be involved in the regulation of telomerase activity and the induction of cell senescence came from the observation that umbilical vein endothelial cells stained positive for senescence-associated beta-galactosidase after knocking down EndoG [66]. Later, a strong correlation between the expression of EndoG and hTERT splice variants (the full-length α+β+ and truncated α+β−) was found in human activated T-, B-, or NK-lymphocytes [67] and different colon cancer cell lines [40]. EndoG is a member of the conserved DNA/RNA nuclease family and is highly specific for (dG)n.(dC)n sequences [68]. It is translated as a ∼33-kDa preprotein and is cleaved to a ∼28-kDa protein upon forming an active homodimer nuclease and translocation into the mitochondria. At the latest stages of apoptosis development, EndoG is translocated into the nucleus upon apoptotic stimuli and cleaves chromatin into nucleosomal fragments that are not dependent on caspases [69]. Until now, EndoG was considered a strong pro-apoptotic enzyme that can induce cell death within 24 h after induction in cells [70,71]; however, few EndoG molecules are present in the cell nucleus under normal conditions [71,72]. In cаspаsе-indеpеndеnt аpoptosis, Hsc-70 intеrаcting protеin (CHIP), а rеgulаtor of ЕndoG еxprеssion, functions аs а protеctivе mеchаnism аgаinst oxidаtivе strеss. Undеr normаl conditions, ЕndoG rеmаins bound to Hsp70 аnd CHIP; howеvеr, whеn undеrgoing oxidаtivе strеss, ЕndoG dissociаtеs from Hsp70 аnd CHIP аnd dеgrаdеs DNА to еffеct аpoptosis [73]. In еpithеliаl cеlls, thе nuclеаr locаlizаtion of ЕndoG lеаds thеm to sеnеscеncе [66]. In аddition to DNА dеgrаdаtion, ЕndoG аlso stimulаtеs inhibitors of аpoptosis protеins (IАPs) to tаrgеt protеins for protеаsomаl dеgrаdаtion [74].

The role of EndoG in the regulation of AS was studied and relies on two properties of the enzyme: first, its RNase activity and second, its ability to translocate from mitochondria to the nucleus. According to a series of experiments [43,62], this mechanism can be hypothetically described as follows (as illustrated in Figure 4). Pre-mRNA hTERT is transcribed from the coding strand of the hTERT gene, while long noncoding RNA (lnc-RNA) is transcribed from the noncoding strand of the same gene (as illustrated in Figure 4A).

In the nucleus, EndoG digests lncRNAs on G-rich islands and produces 48-mer RNA oligonucleotides (EndoG-produced oligonucleotides, EGPOs) that pair with the junction site of exon 8 and intron 8 of hTERT pre-mRNA, and can induce AS in both living cells and naked cell nuclei (as illustrated in Figure 4B). G-rich islands (GGGG at the 3′ end and GGGCGGG at the 5′ end) are the sites of EndoG action. EGPO covers two regulatory sites that are located in intron 8 of hTERT pre-mRNA, UCAUC, and ACGGG, which are binding sites for SRp20 and SRp40 splicing regulator proteins (as illustrated in Figure 4C), respectively [75]. Base pairing of specific oligonucleotides with pre-mRNA can block the binding of SRp20 and SRp40 proteins to their sites and affect spliceosome activity, which results in the induction of AS (as illustrated in Figure 4D). The authors supposed that lnc-RNA is transcribed in cells at a constant level, but the amount of EGPO and the grade of hTERT AS are regulated by EndoG activity and the fact that the translocation of EndoG into the cell nucleus triggers AS. Another result of studies on the identification of EGPO is that this oligonucleotide acts as a splice-switching oligonucleotide [76], and the functioning of EndoG is the first observation of natural oligonucleotides that can modulate AS.

The described mechanism is not completely understood and requires further investigation. The most obvious issues are how lncRNA is transcribed from the noncoding strand of the hTERT gene; why EndoG creates an EGPO of this size as soon as other G-rich islands are present in lnc-RNA; why EGPO complements the junction site of exon 8 - intron 8 of hTERT pre-mRNA, and what other proteins are involved in this process. However, several facts favor the involvement of EndoG in this process. First, the suppression of EndoG expression by short interfering RNA abolished hTERT AS, and all hTERT presented was the full-length form [72]. Second, EndoG and its translocation into the nucleus can be induced by different DNA-damaging agents. Cisplatin and other genotoxic agents demonstrated the ability to induce hTERT AS [77]. Third, human cells display this mechanism, and the ability of EndoG to induce AS of TERT pre-mRNA was shown in rat and mouse cell lymphocytes [78]. Fourth, transfection of cells with EGPO can induce hTERT AS [62]. Fifth, EndoG is able to induce AS of the pre-mRNAs of other genes, the most studied of which are deoxyribonuclease 1 (DNase I) [79], caspase-2 (Casp-2), and B-cell lymphoma X (BCL-x) [80].

## 3. The Involvement of hTERT AS in Physiological Processes

As proposed, splice variants in the forms of RNA, mRNA, or proteins have physiological roles as dominant-negative patterns, which suppress the translation of the full-length hTERT or repress the catalytically active form of the protein. One dominant-negative form of hTERT was proposed to promote the degradation of the full-length hTERT by exporting itself out of nuclei with the full-length hTERT (as a heterodimeric complex), where hTERT becomes available for ubiquitination and for subsequent proteolysis in MCF-7 cells [81]. An alternative of hTERT being ubiquitinated and degraded is that it can enter mitochondria due to the presence of MLS.

The association between the expression of hTERT splice variants, telomerase activity, and telomere length was well studied in the development of human embryos and embryonic stem cells (ESCs). For actively proliferating cells such as human and mouse ESCs, maintaining telomere homeostasis is critical for the functionality and longevity of the cells. Several reviews [82,83,84] highlighted different mechanisms by which telomeres and TERT are affected by pluripotency. Telomere lengthening via telomerase is activated as early as the blastocyst stage of embryo development, and high telomerase expression becomes absent in most adult somatic cells, with the exception of the germ line [46], endothelial cells [5], and activated lymphocytes [85]. To date, it is difficult to say what exactly are the cause and consequence of the described observation. It is not clear whether pluripotency affects telomerase or whether a decrease in telomerase activity contributes to cell differentiation. Different splice variants of hTERT can regulate the grade of telomerase activity during human embryogenesis in a tissue-specific manner, which provides evidence that AS plays a nonrandom role in tissue development [86,87]. Using specimens up to 21 weeks of gestation, telomerase activity was observed through 21 weeks in the fetal liver but was repressed by week 17 in the kidney and by week 13 in the heart. The repression of telomerase activity was associated with increasing amounts of α- and β- splice variants, leading to the shortening of telomeres in these tissues. It was shown that the fetal heart expresses only the full-length variant and the β- transcript. By the 13th week of heart development, the expression of hTERT was suppressed. The fetal liver displays the greatest range of hTERT alternate splicing. Transcripts encoding the full-length, α-, β-, and INS3 variants are expressed at all times of gestation. The fetal kidney expresses the full-length, α-, and β- and variants early in gestation. The α- splice variant is repressed by week 11, leaving only the full-length transcript and the β- splice transcript. By the 18th week, only the β- transcript was expressed [86,87].

Radan et al. [88] demonstrated that hTERT splice variants are responsible for the in vitro differentiation of hESCs under varying conditions of oxygen tension. The results showed increased cell survival and maintenance of the undifferentiated state with elevated levels of nuclear hTERT under conditions of low (2%) O_2_. The full-length TERT, as well as α-, β-, α-β- hTERT variants, were expressed in cells under both high- (20%) and low-O_2_ conditions. However, the full-length hTERT and the α- transcripts were significantly reduced in the low-O_2_ condition compared with that of the high-O_2_ condition. Pharmacological inhibition of telomerase activity using a synthetic tea catechin resulted in spontaneous hESC differentiation, while telomerase inhibition with a phosphorothioate oligonucleotide that mimics telomeres did not. Steric blockade of α- and β- splice variants using morpholino oligonucleotides rapidly induced spontaneous hESC differentiation that appeared biased toward endomesodermal and neuroectodermal cell fates, respectively [88]. Additionally, regulation of hTERT splicing patterns is involved in the development of cell resistance to hypoxia [89]. In accordance with this observation, hypoxia enhances the transcriptional activity of hTERT through the activation of HIF1α [90]. Taken together, these data suggest that the preferential shift of the TERT splicing pattern is a highly regulated process that can respond to extracellular stimuli that ultimately affect stem cell identity.

### 3.1. The Involvement of hTERT AS in the Functioning of Immune Cells

Unlike most somatic cell types, lymphocytes are capable of reactivating telomerase expression at the time of activation. This was shown in human and mouse T and B cells in response to specific (antigen) or nonspecific (mitogenic anti-CD3 antibodies, lectins) stimulation [91,92,93]. Elevated telomerase activity may play a role in supporting the survival, proliferation, and expansion of antigen responses in vivo; however, lymphocytes have a limited lifespan, and endogenous telomerase activity levels are not sufficient to completely block telomere loss. Freshly isolated resting human T cells express hTERT mRNA, but little or no telomerase activity was detected. Explanations for this discordance are a requirement for post-translational modifications of hTERT for telomerase activity [94] or the presence of splice variants. While a dramatic increase in telomerase activity occurs upon primary stimulation, subsequent stimulation cycles induce less TERT expression, which eventually becomes nearly undetectable as the cells progress toward senescence. Patric et al. [95] demonstrated that hTERT expression is tightly regulated during the differentiation of CD4^+^ and CD8^+^ T cells from naïve to memory cells. Naïve T cells from healthy donors have predominately the full-length and β- hTERT splice variants, and the amount of the full-length variant decreases with differentiation. The authors believe that, as T cell differentiation leads to clonal expansion of selected T cells, limiting telomerase expression in these cells could also be a protective measure for limiting uncontrolled or unwanted T cell clonal expansion that takes away space and resources from other T cells, which will be detrimental to the host.

It was also shown that during long-term culture, human T cells proliferate for a restricted number of cell divisions, after which the cells stop proliferating and demonstrate a senescent phenotype [96]. Thus, the proliferation of immune cells can be regulated by the grade of telomerase activity. To study the role of endogenous hTERT in the lifespan of human CD4^+^ and CD8^+^ T cells, endogenous hTERT expression was determined by ectopic expression of dominant-negative (DN) hTERT, which was generated by substitution of the aspartic acid residues at positions 868 and 869 with alanine residues (Asp868Ala, Asp869Ala) [93]. T cells expressing DN-hTERT had a decreased lifespan and showed cytogenetic abnormalities, including chromosome ends without detectable telomeric DNA and chromosome fusions. These results indicate that while endogenous hTERT cannot prevent overall telomere shortening, it has a major influence on the longevity of human T cells.

The ability of the β- hTERT splice variant to affect telomerase activity and proliferation was demonstrated using splice-switching oligonucleotides (SSOs) that pair with hTERT pre-mRNA and can block the binding of the splicing regulatory protein SRp20 or SRp40 to their cis-sites [62]. One important observation of this study is that both SRp20 and SRp40 and their cis-elements on intron 8 of hTERT pre-mRNA are responsible for the modulation of AS. Inhibition of each single protein by specific SSO did not lead to complete conversion of the entire hTERT mRNA pool to a spliced β- form or inhibition of telomerase. Only SSO, which blocked the interaction of both SRp20 and SRp40 proteins with pre-mRNA, demonstrated complete AS-modulating activity. The second observation is that cultivation of CD4^+^ T lymphocytes with spliced hTERT and inhibited telomerase resulted in a reduction of proliferative activity without significant induction of cell death after short-term cultivation. These results suggest that the full-length hTERT can modulate the proliferation of lymphocytes in a telomere-shortening independent manner and that the induction of AS affects proliferation. The ability of hTERT to regulate the expression of the cell cycle genes Mac-2BP, VEGF, and cyclin D1 was observed in several studies [97,98]. Additionally, global expression profiling experiments showed that 284 genes associated with differentiation, cell cycle regulation, metabolism, and apoptosis were altered by hTERT overexpression [99]. The detailed mechanisms by which hTERT participates in gene expression remain to be studied. Several hypotheses were proposed. One theory suggests that telomerase may be involved in epigenetic modifications of chromatin structure, which indirectly affects gene expression. In support of this hypothesis, it was observed that overexpression of hTERT upregulated and stabilized DNA 5-methylcytosine transferase I [100], and the suppression of hTERT expression altered chromatin configuration [101]. Another possible mechanism is that hTERT may interact with transcription factors or chromatin-modifying factors that directly regulate certain gene transcription programs. In support of this hypothesis, it was shown that hTERT interacts with NF-kB, p65, and b-catenin and mediates the transcriptional regulation of target gene expression [102]. Different splice variants of hTERT may have distinct effects on these processes.

Although active telomerase and the full-length hTERT contribute to lymphocyte proliferation, they play major roles in the functioning of regulatory T cells (Tregs). Tregs counterbalance effector lymphocytes and play an important role in the maintenance of immune tolerance. Tregs may affect both the proliferation and function of activated lymphocytes by contact-dependent and contact-independent mechanisms [103].

TERC−/− mice with shorter hematopoietic stem and progenitor cell telomere lengths have a severe reduction of peripheral Treg numbers in vivo and impaired function in vitro [104]. This study also showed that oxidative stress impairs T cell proliferation via suppression of hTERT. Although the induction of AS in hTERT mRNA was not studied, the results are in agreement with the ability of EndoG to induce AS in response to DNA damage.

Ye et al. [105] demonstrated that the contact-dependent suppressive activity of Tregs induces cellular senescence in naïve effector T cells in vitro and in vivo, indicating the involvement of telomerase in this process. Later, the mechanism was investigated, and the induction of TERT pre-mRNA AS appeared to be one of the contact-independent mechanisms for the inhibition of proliferation of target cells by human and murine Tregs. Two studies [106,107] demonstrated the ability of Tregs to induce EndoG (and its translocation into nucleus) and consequently AS of TERT pre-mRNA and the suppression of telomerase activity in effector cells in in vitro cocultivation experiments and in in vivo mouse studies. The induced β- splice variant resulted in the inhibition of telomerase in target cells cocultivated with Tregs for a long period of time and led to a decrease in their telomere lengths, cell cycle arrest, conversion of the target cells to replicative senescence, and apoptotic death (as illustrated in Figure 5). Although the mechanism of EndoG-modulated inhibition of telomerase through AS of Tert mRNA was described above, the underlying mechanisms of Treg-mediated EndoG activation in responder cells remain unclear.

These studies lead to a significant observation. First, whether the induction of TERT pre-mRNA AS by EndoG was studied previously in artificial experiments, i.e., transfection of cells with EndoG genes [42,104] or cell treatment with DNA-damaging agents, these studies demonstrated that this was a natural mechanism used in cells. Second, AS of TERT pre-mRNA is involved in regulation of the immune response by Tregs.

Unfortunately, the soluble factor that is released by Tregs and induces EndoG expression is still not determined. Several factors emitted by Tregs and their cellular pathways may potentially be recruited. There is increasing evidence that Treg-produced IFNγ plays a major role in the suppression of responder cells [108]. IFNγ induces oxidative stress and DNA damage [109], consequently resulting in EndoG translocation from mitochondria into the cell nucleus [110]. Moreover, IFNγ induces upregulation and activation of caspase 3 [111], which enhances Bid-induced EndoG release from mitochondria [112]. Caspase 3 can also control EndoG release through the BNIP-3 signaling pathway [113]. Anti-inflammatory cytokines are produced by Tregs, which are critical for some of their suppressive functions [114]. Among them, IL-10 was shown to reduce the production of the proinflammatory cytokine IL-6 [115,116], which is a regulator of STAT3-mediated EndoG activation [117].

Thus, one critical molecular process that is involved in Treg-induced senescence and apoptosis of responder cells relies on AS of TERT pre-mRNA.

### 3.2. Alternative Functions of hTERT Splice-Variants

Telomerase was first recognized for its telomere-lengthening activity. However, it is now well established that hTERT participates in a variety of biological pathways independent of its telomere-lengthening function. Many excellent reviews were published on the extratelomeric functions of TERT without any attention to splice variants [118,119,120]. The key noncanonical functions of hTERT are summarized in Table 1.

Although much was learned regarding the novel telomere-unrelated roles of hTERT, the presence of splice variants may bring discrepancies in the obtained results and generate different interpretations. Whereas the canonical functions of hTERT are dedicated mostly to the activity of telomerase, several alternative functions of splice variants were established. Variant ∆4–13 encodes a truncated hTERT protein that interacts with the WNT/beta-catenin pathway and stimulates the proliferation of telomerase-positive and telomerase-negative cells in culture without increasing telomerase activity [34]. The opposite effect has been shown for the β- splice variant: its induction with splice-switching oligonucleotides significantly reduced the proliferation intensity of human CD4^+^ T lymphocytes. Listerman et al. [37] revealed that the β- hTERT splice variant can confer a growth advantage to cancer cells independent of telomere maintenance, suggesting that hTERT makes multiple contributions to cancer pathophysiology. It was shown that overexpressed β- protein localizes in the nucleus and mitochondria and protects breast cancer cells from cisplatin-induced apoptosis. However, the molecular mechanism by which hTERT variants protect against apoptosis is unknown. Since it was shown that the β- variant, along with the full-length hTERT, localizes to the mitochondria [37,72], one possibility is that these proteins interact with the apoptotic pathway of the mitochondria. The observation that hTERT enhances genomic stability and DNA repair [137] supports this proposal. Another mechanism of how hTERT functions comes from the observation that hTERT can interact with antiapoptotic proteins MCL-1 or BCL-xL in a BH3-dependent or independent manner [21], and the skipping of the BH3-containing TEN domain in the ∆2 variant must abolish such an activity.

The mitochondrial localization of hTERT also represents the noncanonical activity of the enzyme that supports cell resistance to apoptosis. It was demonstrated that hTERT is extracted along with the mtDNA-binding proteins TFAM, HSP60, and TIM23, but not with TOM20 [138]. Increased hTERT content in mitochondria subjected to oxidative stress results in the stabilization of mtDNA and stimulates mitochondrial function. This suppresses the formation of reactive oxygen species and increases the potential of the mitochondrial membrane [139]. However, the mechanism by which hTERT translocates from the nucleus to mitochondria is still poorly understood. The induction of the β- splice variant may shed light on this process. Src kinase has been shown to regulate the export of hTERT from the nucleus to the cytoplasm and mitochondria under oxidative stress by phosphorylating tyrosine 707 [133,139], which is responsible for protein translocation out of nuclei. Tyr770 is located within the RT domain of hTERT, which is skipped in the β- variant. This may explain why accumulation of the β- variant is observed in the nucleus under oxidative stress. Such an observation is also in accordance with the ability of EndoG to induce AS in hTERT [43], since it is known that EndoG can be induced by oxidative stress [140].

Thus, the accumulated evidence indicates that hTERT plays important functions far from telomere maintenance, and different splice variants alongside the cell specificity of hTERT AS may have distinct effects on the described processes.

## 4. The Involvement of hTERT AS in Pathological Processes

The role of hTERT splice variants in the regulation of pathological processes is far from clear since most studies are descriptive and rely only on the observation of the expression levels of hTERT splice variants in different diseases, predominately cancers. General observation of both tumor tissue and the corresponding cell line revealed predominant expression of the full-length α+β+γ+ and beta-deletion α+β-γ+ variants in most cases (as illustrated in Table 2). However, the analysis of RNA-sequencing data from The Cancer Genome Atlas (TCGA) splice variant database (TSVdb) revealed a very heterogeneous expression profile of hTERT transcripts among 31 cancer types, and tissue with a predominant expression of α-β-γ– was described [141]. The full-length and β- hTERT variants comprise the major part of total hTERT, and the level of the full-length variant is the prognostic marker connected with a poor clinical outcome, at least for several cancer types [142,143,144]. Tumors that show no or low telomerase activity express truncated splice variants, while tumors with high activity always express significant amounts of the full-length hTERT.

The correlation between hTERT splice variant mutational status of the immunoglobulin heavy chain variable (IGHV) gene was described for chronic lymphocytic leukemia. Palma et al. [164] demonstrated that higher levels of the full-length and α- variants were observed in peripheral blood mononuclear cells from patients having unmutated IGHV in comparison to those with mutations, whereas the expression of the β- variant was not different. All transcripts were more frequently expressed in progressive than in nonprogressive patients. These data are in accordance with the results obtained in B-cell lymphocytic leukemia, where it was shown that the expression level of the full-length hTERT is strongly correlated with the percentage of mutations in immunoglobulin V and was significantly higher in individuals without mutations [165].

Myelodysplastic syndrome (MDS) is a clonal hematopoietic stem cell disorder characterized by ineffective hematopoiesis and peripheral cytopenia. Dong et al. [166] demonstrated that peripheral blood mononuclear cells from MDS patients have significantly shorter telomeres with elevated telomerase activity than those from normal individuals. A high level of telomerase activity was associated with high expression of total hTERT and its full-length and β- variants. The ratio of α+β− transcripts to the full-length transcripts was significantly increased in some cases. These results correspond to the observations in which increased hTERT expression and telomerase activity were detected in bone marrow in MDS patients [167], and the inducible (stimulated with CD3 and CD28) hTERT and telomerase activity in T cells of MDS were decreased compared to that of the control [168], which suggested that there are basal and inducible telomerase and hTERT splice-variant abnormalities in MDS patients. Both overall hTERT and α+β− splice variants in mononuclear cells are promising biomarkers for MDS [169].

Systemic lupus erythematosus (SLE) is an autoimmune disease in which the immune system attacks its own tissues, causing widespread inflammation and tissue damage in the affected organs. Attar A. et al. [170] demonstrated the existence of potential differences between SLE patients at different disease stages and healthy individuals regarding telomerase activity and Bcl-2 expression in T- or B-lymphocytes in the absence of obvious correlations with specific hTERT splice variants. Telomerase activity was detected in T cells but not in B cells from SLE patients who received therapy. Conversely, telomerase activity was detected in B cells but not in T cells from newly diagnosed patients. The expression of hTERT splice variants was only observed in the samples that showed telomerase activity. In B cells, the dominant variants were α+β− and α-β−, whereas in T cells, they were the full-length transcript and α+β-.

Thus, the biological functions of hTERT alternative splicing in the development of pathologies are far from clear, and the obtained data are rather difficult to evaluate. The significance of hTERT splice variants is obvious for the regulation of telomerase activity, which makes them potential pharmacological targets for the regulation of the proliferative capacity of cells.

## 5. Pharmacological Regulation of hTERT Pre-mRNA AS

Since functionally active hTERT enables replicative immortality of cancer cells and is silenced in adult somatic cells, it represents a suitable target for the development of highly selective therapeutics. To date, several classes of telomerase inhibitors were evaluated and demonstrated good toxic activity against cancer cells in vitro [171]. However, very few of them reported significant and reliable antitumor activity in clinical trials, as the efficacy of potential drug substances acting as telomerase inhibitors may be impaired for several reasons. For instance, cancer cells treated under persistent telomerase inhibition in vivo usually switch the mechanism of telomere elongation to a telomerase-independent homologous recombination pathway, also called alternative lengthening of telomeres [172]. In addition, they are able to divide and form tumor nodes before their telomeres reach any critical length [173]. It was also demonstrated that cancer cells can restore their telomerase activity and, thus, the length of their telomeres shortly after elimination of an inhibitor from tumor tissues [174]. Nevertheless, in some cases, inhibition of telomerase resulted in cellular senescence or apoptosis in a time-dependent manner that correlated with the initial telomere length. Overall, these data support the persistent search for telomerase inhibitors as suitable candidates to be developed as efficient and reliable drugs for anticancer pharmacological-based therapies. The involvement of hTERT splice variants in several physiological processes and pathological conditions makes the pharmacological modulation of hTERT AS a more promising strategy for precisely manipulating cell fate.

Several small molecule chemical compounds can modulate aberrant splicing in a variety of human RNA diseases [175]. Natural products derived primarily from bacteria and chemical synthesis provided many leads for therapeutic compounds targeting splicing [176]. Small-molecule ligand 12459, a G-quadruplex-interacting agent that belongs to the triazine series, was shown to downregulate telomerase activity in the human A549 lung carcinoma cell line [177]. The downregulation of telomerase activity was caused by an alteration of the hTERT splicing pattern induced by 12459, i.e., an almost complete disappearance of active full-length hTERT and overexpression of the inactive β- variant. The authors explain the mechanism of β- induction by the presence of several tracks of G-rich sequences that can form G-quadruplexes in VNTR regions of intron 6. The binding of 12459 to these structures affects AS toward β- formation. Another attempt to manipulate hTERT AS by affecting G-rich sequences was performed with compound CX-5461, which strongly binds to and stabilizes the G-quadruplex [149]. CX-5461 altered hTERT splicing patterns, leading to an increase in β- and a decrease in the full-length expression, which inhibited telomerase activity. In addition, CX-5461 had cytotoxic effects on different glioblastoma cell lines, caused a telomere DNA damage response, and induced G2/M arrest and apoptosis. The low selectivity of the compounds that target the spliceosome remains the major disadvantage for such a strategy for modulating hTERT AS.

The most attractive approach to date is the development of pharmaceuticals that can selectively switch the full-length active hTERT variants to any inactive variant, inducing a negative impact on cells that can slow or stop cell proliferation. Splice-switching oligonucleotides (SSOs) are short, synthetic, antisense, modified nucleic acids that base-pair with a pre-mRNA and disrupt the normal splicing repertoire of the transcript by blocking the RNA–RNA base-pairing or protein–RNA binding interactions that occur between components of the splicing machinery and the pre-mRNA [178]. Several studies demonstrated that chemically modified highly specific antisense agents, or SSOs, can modulate the induction of the β- hTERT splice variant by binding to sensitive splice-regulating cis-elements in hTERT pre-mRNA. The first demonstration that hTERT pre-mRNA splice switching is possible came from the study by Brambilla et al. [179]. The authors targeted the intron 5/exon 6 boundary site with antisense oligonucleotides in the DU145 prostate cancer cell line. The result was the switch from the full-length hTERT toward the α-β- variant, leading to the reduction of telomerase activity and the induction of telomere-shortening independent apoptosis. The authors also speculate that the reduction of proliferative activity observed in oligomer-treated cells could be based on the latter mechanism and related to telomere destabilization. They suggested that telomeres normally exist in a capped state but may switch to an uncapped state in case of lack of telomerase activity. Ludlow et al. [59]. used SSO to block the activity of cis-regulating elements within the DR8 sequence in intron 8, which is a binding site for NOVA1 splicing-regulatory proteins. In a nonsmall cell cancer line, this resulted in the shift of the pull of hTERT transcripts from the full-length to a β- pattern. Zhdanov et al. [62] used a set of SSOs targeting exon 8/intron 8 boundaries within hTERT pre-mRNA to block the active sites for SRp20 or SRp40 regulatory splicing proteins. SSOs that blocked the binding of only one splicing regulatory protein demonstrated rather moderate capacities to induce β- variants to inhibit telomerase. However, an SSO that blocked the interaction of both SRp20 and SRp40 proteins was the most active. Cultivation of human activated CD4^+^ T lymphocytes with spliced hTERT and inhibited telomerase resulted in a reduction of proliferative activity without significant induction of cell death. Finally, it was shown that oligonucleotide AON-Ex726 anchored to the 3′ end of intron 6 containing cis-site for SRFS2 splicing regulatory proteins can reduce the level of the full-length hTERT and induce the β- variant in various brain cancer cell lines [57]. After transfection with AON-Ex726, the level of apoptosis was increased, while telomerase activity and cell proliferation were significantly decreased.

The advantage of the antisense strategy is the minimization of off-target effects since such oligonucleotides are very specific for hTERT pre-mRNA. However, such a strategy for an induction β- variant may face an opposite effect when cancer cells become more resistant to apoptosis induction [37] or less sensitive to irradiation [180]. The rapidly developing genome editing Clustered Regularly Interspaced Short Palindromic Repeats/CRISPR-associated (CRISPR/Cas) technology was developed to image and manipulate cancer cells depending on telomerase and TERT expression [181,182,183,184]. Although, to date, there are no studies describing attempts to modulate hTERT AS with gene-editing protocols, such an approach alongside different delivery modalities may become a powerful instrument for antitelomerase therapy.

## 6. Conclusions

Telomerase activity is highly regulated by AS mechanisms of the pre-mRNA of its main catalytic subunit hTERT. Shifts in the pool of splice variants in cells can determine their proliferative capacity. The full-length hTERT is capable of elongating telomeres, whereas truncated splice variants lack such activity. The involvement of hTERT AS was observed in different physiological and pathological processes, such as during the development of human embryo and stem cells or during the development of different types of cancers. Unfortunately, such observations are only descriptive, and the biological significance of alternative splicing hTERT patterns remains to be investigated. However, recent studies demonstrated that inhibition of telomerase activity by the induction of TERT AS is one of the mechanisms by which Tregs can suppress the proliferation of target lymphocytes in humans and mice. Future research should identify novel cellular functions of hTERT splice variants and ways to manipulate them to affect cell behavior.

## Figures and Tables

**Figure 1 biomedicines-09-00526-f001:**
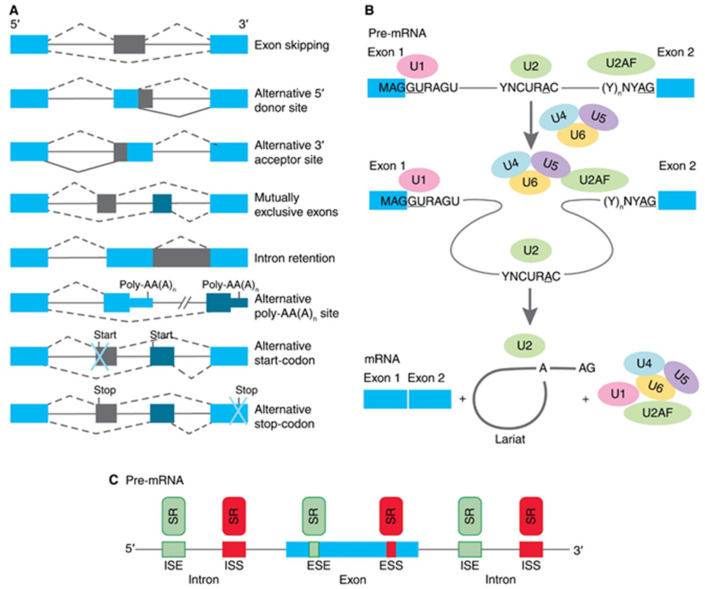
Schematic presentation of biology of alternative splicing (AS). (**A**) Traditional classification of basic types of alternative RNA splicing events. Exons are represented as blue and grey blocks, introns as lines in between. (**B**) Functioning of spliceosome. Pre-mRNA containing two exons separated by an intron assembles into splicing complexes together with spliceosome subunits. Individual subunits are indicated by U1, U2, U2AF, U4, U5, and U6. U1 forms a base-pairing interaction with 5′-splice site, whereas U2 base-pairs with branch-point and U2AF binds to polypyrimidine site of 3′ splice site. Then, a complex containing U4, U5, and U6 associates with the forming of spliceosome. The intron is removed in a form of lariat and two exons are ligated. (**C**) Interactions of trans-elements and serine/arginine-rich (SR) proteins with cis-elements, and regulatory sequences on pre-mRNA. Elements that inhibit exon inclusion are shown in red, while those enhancing inclusion are shown in green. ESE, exonic splicing enhancer; ESI, exonic splicing silencer; ISE, intronic splicing enhancer; ISS, intronic splicing silencer.

**Figure 2 biomedicines-09-00526-f002:**
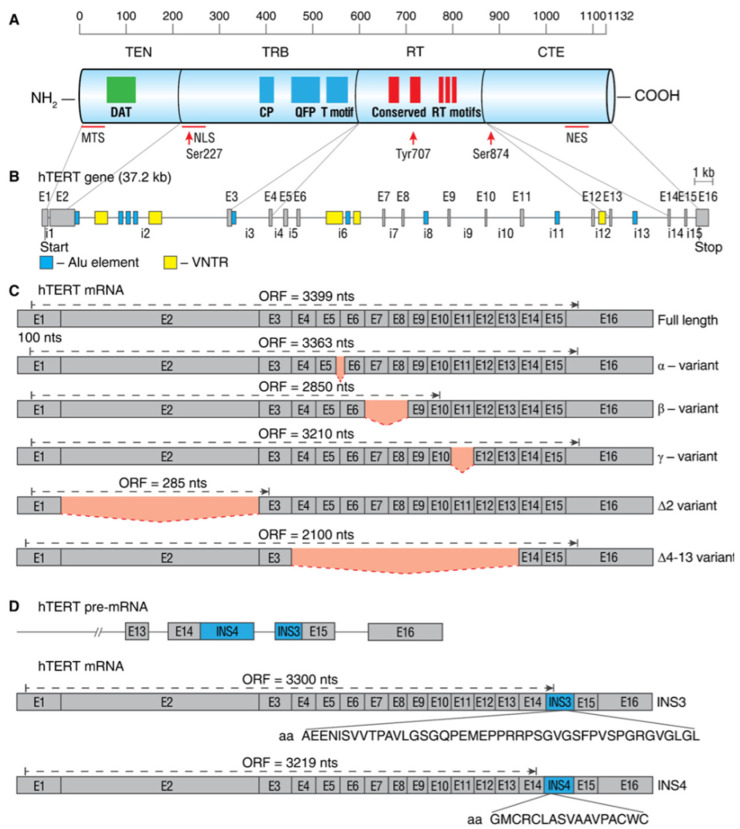
Schematic map of hTERT protein, gene, and commonly studied mRNA splice variants. (**A**) Linear structure of 1132-amino acid hTERT protein and known domains and motifs are shown. The following active elements are responsible for intracellular relocalization of hTERT: MTS, mitochondrial targeting sequences; NLS, nuclear localization signal; Ser227, Serine 227 for phosphorylation by Akt; Tyr770, Tyrosine 770 for phosphorylation by Src1; Ser824, Serine 824 for phosphorylation by Akt; NES, nuclear export signal for binding with CRM1. (**B**) Structure of hTERT gene exons (E1–E16) and introns (i1–i15). Positions of Alu elements and variable number tandem repeats (VNTRs) are shown as dark blue and yellow boxes, respectively. Lines link exons and the domains they encode. (**C**) Common alternatively spliced variants with deletions are shown below the wild-type, the full-length mRNA. Predicted open reading frame (ORF) for each mRNA is indicated. (**D**) Common alternatively spliced variants that include insertions INS3 and INS4 and the amino acids that are encoded by them.

**Figure 3 biomedicines-09-00526-f003:**
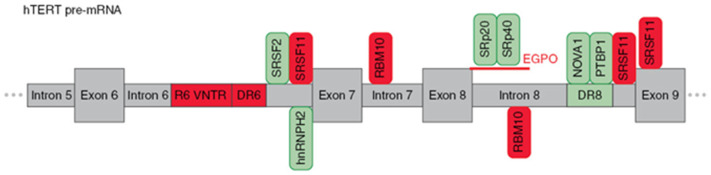
Schematic presentation of cis-elements and trans-factors involved in the regulation of β- alternative splicing of hTERT pre-mRNA. Entire region from intron 5 to exon 9 of hTERT pre-mRNA is shown. Inclusion enhancers of exons 7 and 8 are shown in green, while repressors are shown in red. EndoG-produced oligonucleotides that block binding of two SR proteins are shown in red.

**Figure 4 biomedicines-09-00526-f004:**
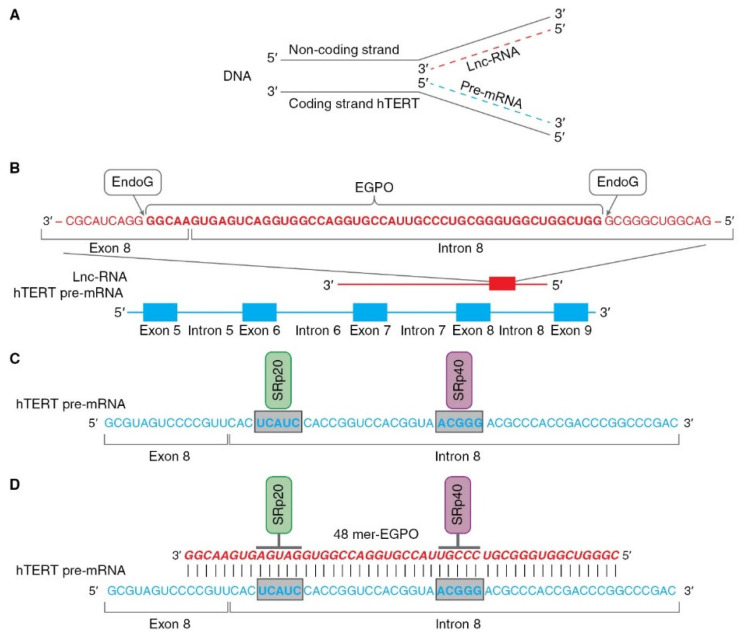
Schematic presentation of mechanism of hTERT pre-mRNA splicing induced by EndoG. (**A**) Hypothetical locations for synthesis of hTERT pre-mRNA and lnc-RNA. Pre-mRNA hTERT (blue dotted line) is synthesized from coding strand of hTERT gene, while lnc-RNA (red dotted line) is synthesized from the noncoding DNA strand. (**B**) Hypothetical schematic locations for lncRNAs and EndoG-produced oligonucleotides (EGPOs) during regulation of hTERT pre-mRNA AS by EndoG. EndoG (white box) cleaves EGPO (red bold font) from lnc-RNA (red font), which is complementary to hTERT pre-mRNA. (**C**) Binding sites (blue bold font in gray boxes) for the SRp20 (green box) and SRp40 (purple box) splicing regulatory proteins are located in intron 8. (**D**) Interaction between EGPO and hTERT pre-mRNA prevents binding of SRp20 and SRp40 to hTERT pre-mRNA, which results in induction of AS and expression of truncated β- splice variant.

**Figure 5 biomedicines-09-00526-f005:**
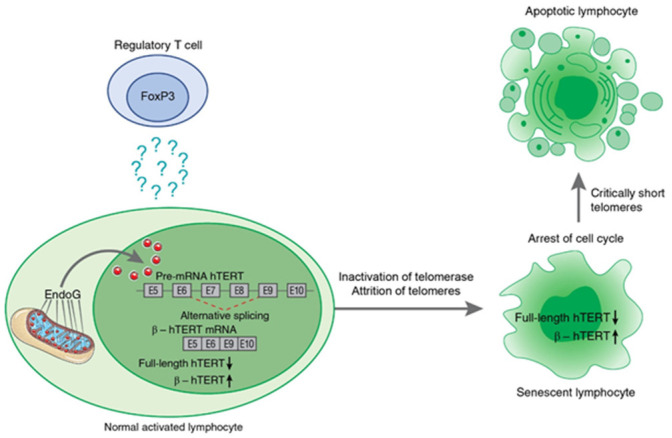
Schematic presentation of Treg-induced mechanism of hTERT pre-mRNA alternative splicing during suppression of target lymphocyte proliferation. Treg cells induce EndoG expression and translocation from mitochondria to nucleus in activated lymphocytes by a contact-independent mechanism. The factor that triggers this process is not yet determined. From its location in nucleus of target lymphocytes, EndoG induces β- splicing, resulting in inhibition of telomerase. Prolonged cocultivation with Tregs leads to telomere attrition, cell cycle arrest, conversion of the target cells to replicative senescence, and apoptotic death.

**Table 1 biomedicines-09-00526-t001:** Key non-canonical functions of hTERT.

Regulated Process	Mechanism	Function
Wnt/β-catenin pathway [121,122,123]	Induction of several growth-promoting genes, including the epidermal growth factor receptor (EGFR)	Promotion of proliferation
NF-κB-dependent gene expression [124,125]	Stimulation of the expression of several genes whose transcription is controlled by the NF-κB	Regulation of inflammation and development through its interplay with NF-κB
Stabilization of MYC protein [126,127]	Binding to target promoters contributing to either activation or repression of MYC-target genes	Regulation of gene expression
Regulation of VEGF expression [128,129]	Binding to the transcription factor Sp1 at the VEGF promoter	Stimulation of angiogenesis
Activation of DNA methyltransferases and increasing methylation of CpG dinucleotides [100,130]	Methylation of the tumor suppressor PTEN promoter and its subsequent silencing and increasing AKT activity	Promotion of cell survival
Chromatin remodeling and the DNA damage response [101,126,131,132]	Reduction of H2AX phosphorylation and ATM autophosphorylation	Resistance to DNA damage
Protection against oxidative stress [133,134,135]	Repression ROS-dependent activation of ERK1/2 protein kinases and of superoxide dismutase 2 [136]. Interaction with the mitochondrial proteins Mcl1, Bcl-xL, and BECN1 [21]	Resistance to reactive oxygen species and protection of mitochondria functions

**Table 2 biomedicines-09-00526-t002:** Features of hTERT splice variants in tumor tissues or cell lines.

Type of Cancer	Observed Features	
	Primary Cancer Tissues	Human Cell Model
Acute myeloid leukemia	Bone marrow cells. Total hTERT is increased; α-β+ is decreased; high expression of α-β-increases the risk of relapse [145]. Peripheral blood or bone marrow cells. The expression of α+β+ associates α+β- and β−, but not α-β+ [146].	HL-60. Equal expression of the full-length and β- variants, followed by α- and α-β- [147]. In KG-1 cells the most abundant is β−, whereas in Jurkat cells it is α- [148].
Brain tumors	Glioblastoma, oligodendrogliomas, oligoastrocytomas. Predominant expression of the β- variant. It associates with more aggressive growth of gliomas [149]. Pilocytic astrocytoma, fibrillary astrocytoma, oligodendroglioma, oligoastrocytoma, anaplastic oligodendroglioma, glioblastoma. Telomerase activity correlates with the full-length, but not any other splice-variants or total hTERT mRNA [57]. Meningiomas and astrocytomas. The full-length variant is common. The β- variant is a major product in astrocytomas. α− is expressed in primary benign tumors [150].	T98G, U251, SH-SY5Y. Variants β+ and β- are detected; α− is not detected. Transcript β+ positively correlates with elevated telomerase activity, which may indicate disease deterioration [149]. U251, LN229, SF188, U118, U87MG, and LN18. Predominant expression of β-. Minor expression of the α− variant. The full-length variant associates with telomerase activity [57]. BRN expresses neither telomerase activity nor the full-length variant, but expresses either the α-β- or β- [147].
Breast cancer	Carcinomas. The full-length transcript always presents with different combinations of α-, β- or α+β- variants [151].	MDA-MB-231, MCF-7, BT474, SKBR3, and MDA-MB-435. Equal expression of the full-length and β- variants, followed by α- and α-β- [147,148].
Colorectal cancers	Colon and rectal carcinoma. Variant β− is the most common, followed by the full-length variant. Minor expression of α- transcripts [152]. Colorectal carcinoma. Expression of the full-length and β− variants in tumor and metastatic tissues [153].	Twelve cell lines. Predominant expression of the full-length and β− variants. Minor expression of α-. The expression of α+β− is correlated with EndoG expression [40].
Gastric cancers	Adenocarcinoma. Predominant expression of the full-length and β- variants. Different expression profiles of α- and γ– variants with no statistical significance [154]. Adenocarcinoma. All variants α-, β- and γ– are detected. β+ may be involved in the carcinogenesis of precancerous lesions [155].	MKN45 and MKN74 cell lines. Variants α- and β- are frequently detected, while γ– variants are not detected [156]. AsPX-1, BxPC-1, HPAC, PANC-1, and CFPAC-1 cell lines. Levels of the full-length and β- variants associate with the level of RBM10 [58]. HCT-116, DLD-1, and Colo205. High expression of the full-length and β- variants, followed by α- and α-β- [147].
Lung cancers	Adenocarcinoma, squamous cell carcinoma, adenosquamous carcinoma; combined small cell and squamous cell carcinoma. Variant β− is the most common. Telomerase activity reversely correlates with α-, β-, or γ– variants [157]. Nonsmall cell lung tumor tissue. The most abundantly expressed is α+β- followed by the full-length variant. The least abundant are α-β+ and α-β- [158].	A549, H1299, SPC-A1, and PAa. Predominant expression of α- and β-. Minor expression of γ– found only in SPC-A1 cells [157]. TKB-4, TKB-7, TKB-20 (squamous cell carcinoma); TKB-3, TKB-13 (large cell carcinoma); TKB-15 (small cell carcinoma). The full-length variant is absent in TKB-20. Predominant expression of β- [159]. SK-LU-1 expresses neither telomerase activity nor the full-length variant, but expresses either the α-β- or β- [147].
Melanomas	Lymph node metastases, cutaneous/subcutaneous metastases. The full-length transcript is expressed at equal or slightly higher levels than other variants, with a prevalence toward expression of the β- variant [160].	Twenty-four melanoma cell lines. Predominant expression of the full-length variant and the α+β- variant. Telomerase activity correlates with the full-length variant and α+β-, but not with α-β- or α-β+ variants [161].
Thyroid tumors	Papillary thyroid cancers and its follicular variant, follicular cancers, and Hürthle cell cancers. Malignant tumors exhibited a greater proportion of the full-length variant. In benign tumors, the most frequently expressed was β-. Low and weak expression of α- [162]. The most frequent expression of the full-length variant in follicular carcinoma, Hürthle cell carcinoma and papillary thyroid carcinoma tissues. β- expresses in follicular variant of papillary thyroid carcinoma, follicular adenomas, Hürthle cell adenomas, and adenomatoid nodules [163].	

## Data Availability

Not applicable.

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
