# Peer review of "Alternative Splicing of Human Telomerase Reverse Transcriptase (hTERT) and Its Implications in Physiological and Pathological Processes"

_biomedicines, 2021, doi:10.3390/biomedicines9050526_

Round 1

Reviewer 1 Report

This is an excellent review both in content, structure and language summarising the current knowledge on the splicing of hTERT. It just requires some minor changes including a brief introduction into general splicing mechanisms.

There are a few minor issues, mainly formal or language-related that have to be addressed by the authors.

  1. Since not all readers might be familiar with the general process of splicing I strongly encourage the authors to include a brief description of the process and its biological consequences. Perhaps even a scheme would be beneficial. Splice factors are mentioned but a general paragraph about the mechanism of splicing would be greatly desirable.
  2. When citing authors in the text, please only use the surname and do not include initials.
  3. line 34: the statement that hTERT levels regulate telomerase activity (T) only applies to most somatic tissues, while it is different in brain where hTR is downregulated very early during development and thus most likely responsible for the disappearance of TA (Ishaq et al., 2016).
  4. Please be aware that also human endothelial cells have TA.
  5. line 129: I think that should rather state: "Some cells may express..." since proteins are unlikely to have hTERT splice variants.
  6. lines 134/5 it is not really obvious what the biological functions have to do with the NMD. Please clarify and perhaps rephrase.
  7. line 149: instead of binding hTERT it should be most likely TR!
  8. line 166: HeLa cells are human cervix carcinoma cells and NOT from mammary gland. Please correct.
  9. line 167:  Involvement of telomerase in cell proliferation can also be via its canonical,telomere-maintaining function. Please clarify and/or rephrase.
  10. Please replace "works" with "studies" throughout the review since the former is not good English.
  11. line 188: When the RNAs with nonsense codons are translated, could that be due to recessive suppressors (e.g. ribosomal proteins as described by the Gawrilows previously)?
  12. The sentence spanning from 213-217 seems to long and complex. better split at least into 2.
  13. line 280: when stating "modulating hTERT" what exactly do you mean here? Expression levels, splice patterns, TA etc? Please specify.
  14. It is not entirely clear what, except for apoptosis, might drive EndoG into the nucleus.Please describe in more detail since this seems important.
  15. line 352: an alternative of hTERT being ubiquitinated and degraded it can enter mitochondria due to its MLS.
  16. line 359: Why would pluripotency affect TERT and telomeres-could it not be the other way around?
  17. lines 379 and 380: what is the difference between "full length TERT" and "hTERT"? Please correct and amend.
  18. lines 398, 406 and others: Please be aware that there is not such a thing as "hTERT activity" (unless you define this specifically), since telomerase activity requires both hTERT and hTR. Please clarify and amend.
  19. lines 438/9: again, similar to above: what is the evidence of non-telomeric functions of hTERT for proliferation? How can you exclude it is via TA?
  20. table 1 referring to ox. stress: at least ref  124 also shows a higher resistance against apoptosis as does ref 129 from the same group. Perhaps worth to mention.
  21. line 581: What doe you mean with "mutated patient"? Please name the mutation which is not applicable for a patient, so please rephrase.
  22. line 671: Again, if TA is decreased, how do you know that apoptosis induction was telomere-independent? It could be via telomere shortening depending on the time frame.
  23.  

minor language issues:
line 45: please add "the" in front of full length.

line 51: instead of "rate" better use "grade or strength"

line 216: replace "in couple" with "in tandem"

line 338: please add "was" in between "presented and full-length"

line 510: please remove space and dot within "Alternative"

line 565: "Level of pattern" is not correct as a pattern cannot have a quantitative level, but is rather a qualitative feature. please rephrase what exactly you mean.

line 669: rather use "switch from" than "switch of"

line 681: reduction "of" not "in"

line 683: it should be "site", not "cite"

line 702 and other sited throughout the ms: please use "splice variants" instead of "splicing variants.

line 706: please add "of hTERT" after "SPLICE pattern"

Author Response

We appreciate the reviewer’s positive responses, including characterizing the paper as “an excellent review both in content, structure and language summarizing the current knowledge on the splicing of hTERT”. In the following response, we address each calling for changes, indicating where relevant corrections have been added to the body of the manuscript and their locations. For reviewers’ convenience all corrections in the text are shown in red color.

We thank to all reviewers for their comments and suggestions.

Response to the comments from the Reviewer #1

Comment 1

Since not all readers might be familiar with the general process of splicing I strongly encourage the authors to include a brief description of the process and its biological consequences. Perhaps even a scheme would be beneficial. Splice factors are mentioned but a general paragraph about the mechanism of splicing would be greatly desirable.

Response:

We included the description of the main processes of alternative splicing in introduction section. One more figure is added.

Comment 2:

When citing authors in the text, please only use the surname and do not include initials.

Response:

Corrected everywhere across the manuscript.

Comment 3:

line 34: the statement that hTERT levels regulate telomerase activity (T) only applies to most somatic tissues, while it is different in brain where hTR is downregulated very early during development and thus most likely responsible for the disappearance of TA (Ishaq et al., 2016).

Response:

Thank you very much for this proposal. We included this statement to the manuscript.

Comment 4:

Please be aware that also human endothelial cells have TA.

Response:

The reviewer is right. The information about endothelial cells and telomerase is added to introduction section.

Comment 5:

line 129: I think that should rather state: "Some cells may express..." since proteins are unlikely to have hTERT splice variants.

Response:

Off course they are cells. Corrected. Thank you!

Comment 6:

lines 134/5 it is not really obvious what the biological functions have to do with the NMD. Please clarify and perhaps rephrase.

Response:

The sentence was rephrased for “The biological significance of these variants for cells remains to be investigated, as they must be degraded by nonsense-mediated decay.”

Comment 7

line 149: instead of binding hTERT it should be most likely TR!

Response:

The reviewer is right. It is hTR. Corrected.

Comment 8

line 166: HeLa cells are human cervix carcinoma cells and NOT from mammary gland. Please correct.

Response:

It is right. Corrected. Thank you!

Comment 9.

line 167:  Involvement of telomerase in cell proliferation can also be via its canonical,telomere-maintaining function. Please clarify and/or rephrase.

Response:

The sentence is modified for better clarification.

Comment 10

Please replace "works" with "studies" throughout the review since the former is not good English.

Response:

Corrected everywhere across the manuscript.

Comment 11

line 188: When the RNAs with nonsense codons are translated, could that be due to recessive suppressors (e.g. ribosomal proteins as described by the Gawrilows previously)?

Response:

We are very sorry, but we could not find any paper published by the author Gawrilows. Most of the papers related to ribosomal recessive suppressors published in 1980th with closed access. The answer for this question would be very speculative, so we would prefer not to include it to the manuscript. If the reviewer feel that this issue is really important for the integrity of the work, we are kindly asking to provide us more information about this phenomenon.

Comment 12

The sentence spanning from 213-217 seems to long and complex. better split at least into 2

Response:

Corrected according to recommendation.

Comment 13

line 280: when stating "modulating hTERT" what exactly do you mean here? Expression levels, splice patterns, TA etc? Please specify.

Response:

Corrected to “… which could modulate AS of hTERT pre-mRNA and consequently biological properties of hTERT protein.”

Comment 14

It is not entirely clear what, except for apoptosis, might drive EndoG into the nucleus. Please describe in more detail since this seems important.

Response:

We introduced the relevant information about EndoG to the manuscript.

Comment 15

line 352: an alternative of hTERT being ubiquitinated and degraded it can enter mitochondria due to its MLS.

Response:

Thank you very much for this idea. We included this statement to the manuscript.

Comment 16

line 359: Why would pluripotency affect TERT and telomeres-could it not be the other way around?

Response:

We added two suggestions to highlight this feature. “To date, it is difficult to say what exactly is the cause and what is the consequence of the described observation. It is not clear whether pluripotency affects telomerase or whether a decrease in telomerase activity contributes to cell differentiation.”

Comment 17

lines 379 and 380: what is the difference between "full length TERT" and "hTERT"? Please correct and amend.

Response:

Corrected.

Comment 18

lines 398, 406 and others: Please be aware that there is not such a thing as "hTERT activity" (unless you define this specifically), since telomerase activity requires both hTERT and hTR. Please clarify and amend.

Response:

Yes, you are absolutely right. Corrected to telomerase activity where appropriate.

Comment 19

lines 438/9: again, similar to above: what is the evidence of non-telomeric functions of hTERT for proliferation? How can you exclude it is via TA?

Response:

According to the described experiment, the cultivation of lymphocytes with induced β– variant was performed within 96 h. This time is not enough for telomere attrition and inhibition of cell cycle. The authors of referenced study proposed such an explanation.

We modified the sentence to “… modulate the proliferation of lymphocytes in a telomere-shortening independent manner and that the induction of AS affects proliferation.”

Comment 20

table 1 referring to ox. stress: at least ref  124 also shows a higher resistance against apoptosis as does ref 129 from the same group. Perhaps worth to mention.

Response:

We agree. Corrected.

Comment 21

line 581: What do you mean with "mutated patient"? Please name the mutation which is not applicable for a patient, so please rephrase.

Response:

Corrected

Comment 22

line 671: Again, if TA is decreased, how do you know that apoptosis induction was telomere-independent? It could be via telomere shortening depending on the time frame.

Response:

According to the described experiment, the cultivation of DU-145 cells with R7 active oligomer was performed within 72 h. This time is not enough for telomere attrition. However, authors speculated that the reduction of proliferative activity observed in oligomer-treated cells could be based on the latter mechanism and related to telomere destabilization. They suggested that telomeres normally exist in a capped state but may switch to an uncapped state in case of lack of telomerase activity.

We modified the statement in our manuscript.

Comment

line 45: please add "the" in front of full length.

Response:

Corrected everywhere across the manuscript.

Comment

line 51: instead of "rate" better use "grade or strength"

Response:

Corrected everywhere across the manuscript.

Comment

line 216: replace "in couple" with "in tandem"

Response:

Corrected.

Comment

line 338: please add "was" in between "presented and full-length"

Response:

Corrected

Comment

line 510: please remove space and dot within "Alternative"

Response:

Corrected

Comment

line 565: "Level of pattern" is not correct as a pattern cannot have a quantitative level, but is rather a qualitative feature. please rephrase what exactly you mean.

Response:

We are not sure that we understood this concern on a wright way. The sentence was modified to “… since most studies are descriptive and rely only on the observation of the expression levels of hTERT splice variants in different diseases, predominately cancers.”

Comment

line 669: rather use "switch from" than "switch of"

Response:

Corrected

Comment

line 681: reduction "of" not "in"

Response:

Corrected

Comment

line 683: it should be "site", not "cite"

Response:

Corrected

Comment

line 702 and other sited throughout the ms: please use "splice variants" instead of "splicing variants.

Response:

Corrected everywhere across the manuscript.

Comment

line 706: please add "of hTERT" after "SPLICE pattern"

Response:

Done.

I must notice that this manuscript was edited for proper English language, grammar, punctuation, spelling, and overall style by American Journal Experts editing service.

Once again, we thank the reviewer and the editorial office for thorough review and hope that the corrections have improved this manuscript.

Dmitry D. Zhdanov, Ph.D.

Laboratory of Medical Biotechnology, Institute of Biomedical Chemistry, Pogodinskaya st, 10/8, Moscow, 119121, Russia

zhdanovdd@gmail.com

Reviewer 2 Report

This manuscript provides a complete, extensive and insightful review of the literature on the different splice variants of hTERT and their physiological and pathological functions. The review is well written and overall, I think it’s a useful contribution for the field.

Minor points:

  1. some of the figures have poor resolution and do not show well in the pdf file.
  2. In figure 1, it might be helpful to add relevant references next to the splice variants
  3. line 127: "...is trаnslаtеd intо а truncаtеd prоtеin, which еnаblеs thе fоrmаtiоn оf аctivе tеlоmеrаsе but prоtеcts cеlls frоm  Ð°pоptоsis"  are the authors sure that the truncated protein makes an active telomerase? Looks like a "not" is missing in this sentence.
  4. line 129: "Sоmе prоtеins mаy еxprеss this β‒ hTЕRT vаriаnt in humаn аctivаtеd CD4+ T lymphоcytеs..." this sentence does not make much sense to me, do the authors mean "some cells" instead of "some proteins"?
  5. Line 165: Saos-2 cells, as far as I know, are telomerase-negative, at least in terms of telomerase activity and maintain telomeres through ALT, similarly to U2OS.
  6. Line 181: there is a typo inside the first reference bracket.
  7. Line 211: not that it matters much, but the blocks in the figure appear orange instead of red.
  8. Line 295: "DNA" can be omitted from the sentence
  9. Line 353: the reference can be moved at the end of the sentence
  10. Line 483: This whole paragraph is a little confusing and might need rephrasing. Maybe removing the brackets and placing a "first" before the sentence that starts with "second"
  11. Line 510: There is a typo in the title.
  12. Line 698: Conclusion-s-

Author Response

We appreciate the reviewers’ positive responses, including characterizing the paper as “review is well written and overall, … it’s a useful contribution for the field”. In the following response, we address each calling for changes, indicating where relevant corrections have been added to the body of the manuscript and their locations. For reviewers’ convenience all corrections in the text are shown in red color.

We thank to all reviewers for their comments and suggestions.

Response to the comments from the Reviewer #2

Comment 1

some of the figures have poor resolution and do not show well in the pdf file.

Response:

We uploaded initial figure files in EPS format. We believe the quality will be fine after final production of the article.

Comment 2:

In figure 1, it might be helpful to add relevant references next to the splice variants

Response:

The relevant references were added to the figure.

Comment 3:

line 127: "...is trаnslаtеd intо а truncаtеd prоtеin, which еnаblеs thе fоrmаtiоn оf аctivе tеlоmеrаsе but prоtеcts cеlls frоm  Ð°pоptоsis"  are the authors sure that the truncated protein makes an active telomerase? Looks like a "not" is missing in this sentence.

Response:

The reviewer is right. Thank you. Corrected.

Comment 4:

line 129: "Sоmе prоtеins mаy еxprеss this β‒ hTЕRT vаriаnt in humаn аctivаtеd CD4+ T lymphоcytеs..." this sentence does not make much sense to me, do the authors mean "some cells" instead of "some proteins"?

Response:

Of course, they are cells. Corrected. Thank you.

Comment 5:

Line 165: Saos-2 cells, as far as I know, are telomerase-negative, at least in terms of telomerase activity and maintain telomeres through ALT, similarly to U2OS.

Response:

The reviewer is right. There is a mistake in the composition of the sentence. Corrected

Comment 6:

Line 181: there is a typo inside the first reference bracket.

Response:

It is not a typo. It means that the beginning of the insertion at 5`-end is not known. Such Ñ„ designation is used by the authors of the article described (Saebøe-Larssen S, Fossberg E, Gaudernack G. Characterization of novel alternative splicing sites in human telomerase reverse transcriptase (hTERT): analysis of expression and mutual correlation in mRNA isoforms from normal and tumour tissues. BMC Mol Biol. 2006 Aug 29;7:26. doi: 10.1186/1471-2199-7-26.)

No corrections were made.

Comment 7:

Line 211: not that it matters much, but the blocks in the figure appear orange instead of red.

Response:

Corrected for red color.

Comment 8

Line 295: "DNA" can be omitted from the sentence

Response:

Done.

Comment 9

Line 353: the reference can be moved at the end of the sentence

Response:

Done

Comment 10

Line 483: This whole paragraph is a little confusing and might need rephrasing. Maybe removing the brackets and placing a "first" before the sentence that starts with "second"

Response:

Corrected according to recommendation.

Comment 11

Line 510: There is a typo in the title.

Response:

Corrected

Comment 12

Line 698: Conclusion-s-

Response:

Corrected

I must notice that this manuscript was edited for proper English language, grammar, punctuation, spelling, and overall style by American Journal Experts editing service.

Once again, we thank the reviewer and the editorial office for thorough review and hope that the corrections have improved this manuscript.

Dmitry D. Zhdanov, Ph.D.

Laboratory of Medical Biotechnology, Institute of Biomedical Chemistry, Pogodinskaya st, 10/8, Moscow, 119121, Russia

zhdanovdd@gmail.com